# Functional Freedom: A Psychological Model of Freedom in Decision-Making

**DOI:** 10.3390/bs7030041

**Published:** 2017-07-05

**Authors:** Stephan Lau, Anette Hiemisch

**Affiliations:** 1Department of Psychology, Florida State University, 1107 Call Street, Tallahassee, FL 32306, USA; 2Department of Psychology, University of Greifswald, Franz-Mehring-Straße 47, Greifswald D-17487, Germany; hiemisch@uni-greifswald.de

**Keywords:** free will, freedom, decision-making, underdetermination, volition, consciousness

## Abstract

The freedom of a decision is not yet sufficiently described as a psychological variable. We present a model of functional decision freedom that aims to fill that role. The model conceptualizes functional freedom as a capacity of people that varies depending on certain conditions of a decision episode. It denotes an inner capability to consciously shape complex decisions according to one’s own values and needs. Functional freedom depends on three compensatory dimensions: it is greatest when the decision-maker is highly rational, when the structure of the decision is highly underdetermined, and when the decision process is strongly based on conscious thought and reflection. We outline possible research questions, argue for psychological benefits of functional decision freedom, and explicate the model’s implications on current knowledge and research. In conclusion, we show that functional freedom is a scientific variable, permitting an additional psychological foothold in research on freedom, and that is compatible with a deterministic worldview.

## 1. Introduction

For a long time, the reality of free will has been debated within philosophy. Elaborate arguments have been put forward in favor of, as well as against, its compatibility with a deterministic world [1,2,3,4]. Instead of participating in this debate, we presume that psychology can develop its own models of *decision freedom*, without metaphysical ballast, for the purpose of a productive engagement with the topic of human freedom [5,6].

The theory of reactance [7] explains motivational reactions to threats of behavioral freedoms, and is only one example of how psychological research can put the concept of freedom to good use. More recent work connected the concept of free will with various research areas in psychology. In social psychology, it was argued that autonomous and free action regulation is of high value to research on well-being and motivation [8]; that free and less-free actions can be operationalized in terms of capacities such as self-control [9]; and that variations in just the belief in free will already yield impacts on moral behavior [10]. The newly emerging field of neurovolition has begun to view voluntary, volitional actions as distinctive phenomena and to conceptualize the brain mechanisms behind self-generated action [11,12]. Moreover, recent research has demonstrated that folk psychological notions of free will do not seem to revolve around metaphysical assumptions (e.g., a soul or the absence of causal determination), but instead are rooted in beliefs about choice capacities [13,14]. The gist of all these intriguing approaches is that psychology does not need to engage in the metaphysical debate about free will [5], but should go ahead with own naturalistic, operationalizable notions of freedom as a capacity in decision-making [15,16] and action generation [12]. So far, there is no account that explicates all these ideas about capacities or integrates them with decision mechanisms and conditions into a holistic model of decision freedom. We strive to contribute to psychology’s newfound engagement with freedom by outlining such an account.

Our aim is to propose a model that helps to assess the degree of freedom within a concrete decision episode as a parameter. The model is based on psychological theories on decision freedom, as well as philosophical literature on how to conceive a naturalistic free will. We generally echo the view that psychological freedom is rooted in capacity, especially capacities that revolve around decision-making and choice [14,17]. By defining the concept of *functional decision freedom* as dependent on three decision-related dimensions, we are able to describe how and why decisions can be assessed as more free or unfree. In the course of this paper, we will derive and develop these dimensions and their indicators, as well as discuss the adaptive benefits of free decisions and the resulting implications for further research.

## 2. Defining Functional Freedom and Its Scope

Functional freedom is the central variable of the model, and a quantitative parameter pertaining to a decision episode. At the core of a decision episode is always an individual who decides on preferences or higher-order intentions, which should guide subsequent actions in a specific situation. Functional freedom thus describes a type of person–situation interaction in the context of decision-making. In a functionally free decision episode, an individual is able to control, by means of internal processing, how inputs (e.g., social stimuli, behavioral triggers) are mediated, and how the corresponding output (choices, goals, and behavior) is shaped. More specifically, functional freedom arises if the deciding individual is able to re-determine external and internal stimuli to some extent, by employing certain abilities and processes. We will specify these abilities and processes, as well as their situational demands, in the model. Generally speaking, the idea is that functional freedom describes the extent of self-governed action control during decision-making, and represents the capacity to reconfigure inputs and align them to personal needs and values.

Functional freedom is relative, and clearly limited. High functional freedom in a decision is given when the decision process is highly reflective, when the decision’s structure is highly underdetermined and the decider is highly rational and skilled. On the contrary, decision episodes which are functionally unfree would be characterized by automated, fast, and easy decision-making that demands no further skills, rational requirements or conscious values. Importantly, this does *not* entail that the latter decisions are inferior to free decisions, or are to be refrained from. Indeed, most everyday decisions are likely automated and subconscious for good reasons [18,19]. However, the fact that decisions with a potential for high functional freedom are perhaps scarce in daily routine does not render them trivial. Deliberate decision-making, a key component of functional freedom, is typically associated with problematic decisions that bear highly important consequences for our lives.

Some clarification is in order before we develop the dimensions and indicators of functional freedom. There are many—often synonymously used—terms when dealing with human freedom: free will, free agency, inner freedom, volition, voluntary behavior, and so on. We refrain from using these, and stick to the use of decision freedom, because it reflects the central idea: freedom as the capacity to (partly) shape existence by ourselves.

## 3. The Model of Functional Freedom in Decision-Making

### 3.1. Rationale of the Model Construction

There is rich philosophical and psychological literature following the notion that freedom in decision-making represents a psychological capacity. This dates back to Spinoza and his compatibilistic notion of free will through insight [20]. However, the authors following this line of thought differ in the localization of this capacity. Freedom or free will is either attributed to personal dispositions, to process features, or to structural parameters (e.g., number of options). While all these positions have merit, we concluded that a comprehensive model of decision freedom must integrate them. Decisions are complex phenomena that can be (and are) analyzed and investigated on the levels of the process, the individual decision-maker, or the choice structure. As we show below, decision freedom can theoretically vary on all these dimensions, too. We therefore map functional freedom to these three dimensions, while we use the theories of authors focusing on each dimension to define the characteristics that entail either high freedom or high unfreedom.

In order to facilitate high functional freedom in a decision episode, the following conditions must be met: self-reflection, underdetermination, and rationality. We will explicate the underlying ideas and review the literature for these three conditions, as well as explain the dimensional indicators that represent high and low values. Different degrees of overall functional freedom are the consequence of different values on the dimensions. Hence, the functional freedom of a decision episode will be conceptualized as a function whose input is equivalent to the indicator values of the three dimensions (Figure 1).

Subsequently, we will discuss the properties, possible adaptive benefits, and implications of the resulting functional freedom.

### 3.2. Self-Reflection—Freedom of the Decision Process

#### 3.2.1. Idea and Background

Imagine a person that constantly avoids conscious effort and never reflects on his or her actions, while abstaining from thorough evaluations of events, options, actions, and so on. By analogy, imagine a drifter on a lively, interesting marketplace in an exotic country. One moment he or she is drawn to a booth because it looks so colorful; the other moment, a pushing of the crowd determines the direction; another time, the person moves over there because of a nice smell. Although such drifting may be relaxing and enjoyable, it is nevertheless unfree because, to stay in the analogy, the current decides where it takes us [21]. If we let external stimuli and events *directly* influence our behavior, we are per definition lacking functional freedom.

A free decision is associated with conscious thought [22] that refers to the decision and the self—hence “self-reflection”. To be sure, self-reflection is not a necessary feature of consciousness in general. I can be consciously aware of a decision that I make intuitively, without thinking about or my own needs or reflecting on mental events [22,23,24]. Self-reflection is used here as an umbrella term for specific modes of thought that (a) can only occur within consciousness, and (b) entail freedom. This encompasses processes such as mental simulation, prospection of events, thoughts about thoughts, and reflective argumentation. Mapped to dual-process models of information processing [24,25,26], self-reflective conscious thought accords to System 2 activity. System 2 processes are characterized as controlled, symbolic, logical, slow, and requiring of attention, in contrast to System 1 processes, which are fast, automatic, nonverbal, intuitive, and subconscious. Such controlled conscious thinking increases the open-mindedness and critical evaluation of actions, fostering the consideration of abstract values and long-term goals. However, we do *not* claim that free decisions are based solely on conscious thought. Rather, System 1 and System 2 supplement each other in a complex interplay [24,26]. We therefore assume that self-reflective conscious thought must be present to some extent in a decision, and the higher its share in the decision outcome, the higher the functional freedom of the process dimension.

Although some authors reject the necessity of conscious processing for freedom [27], it is a recurrent theme in the literature. For instance, Spinoza connects achievable inner freedom closely to his concept of adequate ideas [20,28]. These are thoughts that are drawn directly from the self through the use of insight. Frankl and Fromm describe decision freedom in terms of appraisal processes consisting of the re-evaluation of incoming stimuli and events [29,30]. According to these authors, we are free in *how to judge* events affectively and morally. This appraisal determines, partly, how the stimuli affect oneself and thus alter the behavioral reaction [29]. In a similar vein, Johnson-Laird describes freedom as a capacity of information processing, specifically, the indirect shaping of the thought–action interface [31]. This shaping could occur by the generation or modification of rules and methods that are used in further judgment and decision-making (e.g., “When I think I know the best option, I try to generate two more arguments for the alternative”). While Tuomela [32] and Bieri [21] refer to inner freedom as the process of generating meta cognitions (e.g., intentions about intentions, ability to judge and monitor one’s own cognition), recent studies likewise associate folk concepts of freedom with conscious processing and volition with the integration of various information qualities [12,33]. Summing up, we include conscious processing as an essential component of functionally free decisions, specifically those processes that alter the direct effect of incoming inputs and stimuli.

#### 3.2.2. Dimensional Indicators

The following processing features characterize high self-reflection and thus a high functional freedom of the decision-making process:(1)A direct and intense focus of attention on the decision problem and its (mental) context.(2)A complex integration of anticipative, knowledge-based, and evaluative information and processes.(3)A high number of generated mental propositions (e.g., thoughts about other mental propositions; new insights and connections between representations; changes in preferences, beliefs or expectancies).(4)A greater activation, modification or generation of decision-related rules.(5)The decision is based strongly on logical and justifiable arguments.


The unfree opposite would be a decision that is made completely subconsciously, with a great reliance on habits, schemas and automated processing. Of course, an intuitive style of decision making is *not* dysfunctional per se. In fact, the majority of everyday decisions are probably made effortlessly and successfully, with little access to consciousness (e.g., menu choices, navigating traffic lights, making waypoint decisions while commuting) [18,19,24]. But such decisions are unfree to a greater extent because unconscious or automated processing is more rigid [34,35,36], reliant on external stimuli, and hardly evaluated by the self, thus more prone to external manipulation. By contrast, conscious reflective processing enables abstract attitudes and values to influence decision outcomes and increase the personal control over direct causal influences through an increased awareness and ability to re-determine them [22,37,38].

### 3.3. Underdetermination—Freedom of the Decision Structure

#### 3.3.1. Idea and Background

If one must choose between a tasty-looking apple and a rotten, shriveled one, the decision is determined by the choice set. Almost everybody would go for the tasty apple. However, imagine you must choose between a weekend on a conference, where you are invited as keynote speaker, and a sailing-trip with a long-missed friend, who is available only on that weekend. In this case, the decision is underdetermined because there is a conflict and it is hard to foresee the decision outcome.

The amount of functional freedom in a decision episode is also specified by the structural aspects of the decision at hand. A decision structure is composed of internal factors, such as the preferences the decider brings into the situation, and of situational factors, such as the size and values of the choice set. As the examples above illustrate, a decision structure can be more or less determined, and thus contain more or less freedom. A free and underdetermined decision is one in which the decider *himself* has to tip the scales, and where the outcome becomes increasingly unpredictable. The presence of motivational conflict (e.g., when preferences are equally served by different options of the choice set) and ambiguous information (e.g., about the valence of options for certain preferences) create such a decision structure. Consequently, such a decision is freer than a determined decision, in which the outcome is almost exclusively prescribed by the situation, for example via a clear dominance structure that triggers a homogenous choice behavior in different people. Several authors advocated that conflict-laden, ambiguous, or unknown decision structures are a key element of free decision-making [27,32,37,39,40,41,42,43], as they entail an epistemic openness of the choice. Please note that underdetermination describes a balance between forces, motives or choice alternatives, not *in*determination.

Psychologically, underdetermined decisions are further characterized by a high subjective uncertainty about the right choice [42,44]. This uncertainty can be understood as an affective mechanism indicating complex and difficult choices that demand more cognitive resources to be adequately solved [45]. It may be uncomfortable, but at the same time helpful, to realize the choice as a relevant opportunity to exert free control. Though underdetermined choices may be rare in everyday life, they will definitely occur. Given the affluence of motives and goals of human beings, conflicts between wants and goals are inevitable. They are also likely to have a high relevance for the individual [18]. Prototypical examples of underdetermined decisions are moral and social dilemmas (e.g., personal benefits vs. altruistic behavior).

The more underdetermined a decision structure is, the less likely there is a default choice or dominant option. Hence, the decision-maker must reframe the options, and construct a preference by himself. Self-reflection and rationality have to come in to resolve underdetermined decisions in accordance with one’s values and needs, thereby enabling self-determination and thus a high degree of freedom. This implies that the dimension of underdetermination is not independent from the other dimensions, an issue we will discuss later.

#### 3.3.2. Dimensional Indicators

The following characteristics indicate higher underdetermination and therefore higher functional freedom of the decision structure:(1)There are many relevant options.(2)There is a conflict between options (i.e., wants, motives) because they are mutually exclusive or very similar in value, so that the difference in preferences is very small.(3)The decision situation is unknown and new, so that there is no routine or default option available.(4)The decision outcome is of relevance for the individual and the self, as trivial choices (e.g., “which of two full bottles of water should I take?”) are likely to be skipped, for example, by picking randomly.(5)A larger degree of ambivalence within each option (i.e., option features with positive vs. negative valence).


By contrast, determination in the decision structure increases when there is only low choice (for instance, few options with low values available to the decision-maker); when there is a dominant option, which is chosen with a high probability and without further consideration; or, when the decision situation is well-known, allowing the successful implementation of behavioral routines or default options. These characteristics indicate a decrease of freedom in the decision’s structure.

Apparently, determined decisions, such as the tasty apple, are more predictable and therefore predestined to a certain degree. Empirically, this would result in large differences in the frequency distributions of choices (e.g., 100% vs. 0%). On the contrary, underdetermined decisions are rather unpredictable, as in the sailing vs. keynote dilemma. The outcome would be dynamically influenced by the numerous and fluctuating variables of the decision-maker’s personality and inner processes. The dispositions involved are highly interdependent and recursive, difficult to assess, and therefore difficult to predict in the long run [37]. Hence, the frequency distribution of an underdetermined choice should approach a balance, reflecting the equilibrium of values.

### 3.4. Rationality—Freedom of the Decider

#### 3.4.1. Idea and Background

Consider a person that suffers from a severe disorder to control her fears. Each time the person enters a potentially threatening situation (such as an exam), she panics and flees the room. It is unlikely that you would hold the person responsible for this, after observing this pattern several times. More likely, we would like to help, for example by suggesting an intervention such as behavioral therapy. If this intervention is successful, the person would learn to endure the fear and develop behavioral alternatives. Afterwards, she would be able to do things she could not do before: staying put and answering exam questions, for instance. By this growth in ability, the person is able to do and achieve things she *wants*.

The point here is: in order to consciously generate free intentions and actions in conflict-laden, ambiguous decisions, the individual must be able to do so. Rationality thus comprises the personal requirements of functional freedom (Figure 1). We use the term “rationality” in the sense of reason, that is, traits and abilities associated with decisions and goal attainment that follow from enlightened self-interest, and help to reach states that are good for us [46] (p. 5, for a similar definition). Moreover, rationality represents an essential boundary condition for responsibility [47], which is seen as a common correlate and consequence of freedom.

The rationality aspect of functional freedom comprises two important components: (a) abilities, and (b) a system of motives and values.

Which abilities do matter in respect to freedom? Generally, all self-regulatory and cognitive mechanisms do matter that help to overcome impulses and hard-wired routines; endure uncertainty and ambiguity; systematically direct conscious thought; and enable the generation of new rules, ideas or perspectives. In line with this, many authors place an emphasis on personal capacities when they describe a natural free will. Walter and Bieri describe self-reflection in part also as a dispositional competence that can vary across individuals [21,48]. Similarly, Spinoza associates adequate ideas with the dispositions of virtue and intelligence [20]. Baumeister argues for a cluster of abilities (i.e., self-control, rational choice, planning, and initiative) that enable free will psychologically [9,49]. According to Baumeister, these abilities increase cultural adaptability by decreasing the influence of external stimuli and habits in social behavior, thereby gaining degrees of freedom. In agreement, Tuomela describes freedom as the ability to prioritize second-order propositions (i.e., intentions directed at other mental propositions, such as desires) over natural habits (e.g., the desire to drink alcohol) [32]. The ability to control impulsive behavior in favor of rational long-term goals thus represents a crucial theoretical component of personal freedom. Finally, decision freedom also incorporates the ability to learn from decisions and behavioral outcomes, leading to changes in the capacity to decide freely [30,41,42]. This implies that on the individual-level, functional freedom is responsive to cognitive and self-regulative enhancement (e.g., insights, positive learning experiences). The indicators of high rationality resulting from these ideas can be found in the next section.

In addition to abilities, rationality must also comprise a system of personal values. This system delivers the goals for rational thinking in the first place [46], by containing ethical beliefs or personal attitudes. Without it, conscious deliberation would have no basis and freedom of will would be pointless. We believe this assumption is vital to any conception of decision freedom because it brings the personality into play. Following Tuomela and Baumeister [9,32], if we removed long-term (life-)goals and ethical attitudes from decision-making, we would not be in need of a freedom capacity. Intuitions would simply suffice.

Which values exactly inform this system is determined by a person’s background. Accordingly, if a value system is informed by secular humanistic ideals, it might well lead to decisions that are commonly seen as ethical and “good”. It could, however, also foster acts that are selfish, destructive, or otherwise, depending on the ethos of the person.

Essentially, a rational and free decision-maker *must* possess a guiding value system, accessible via consciousness and reflection. On the one hand, this can lead to positive ethical and moral decisions if the condition of a deeply humanistic value system is given [21,30]. On the other hand, this condition is not mandatory. A person entertaining a misanthropic value system (e.g., a sadistic dictator) could decide functionally freely, according to his values, as well.

#### 3.4.2. Dimensional Indicators

The following dispositions and abilities indicate higher rationality (i.e., in the sense of reasoned thinking and behavior) and therefore higher functional freedom:(1)High self-regulatory skills, such as self-control and emotion regulation, as well as high cognitive abilities, such as creative problem-solving [21] or enhanced working memory capacity [50].(2)Self-awareness and self-knowledge, as well as knowledge about the world—this might be subsumed under the trait of wisdom (i.e., a generalized experience with and understanding of world affairs) [51].(3)Functional, intact memory processes (e.g., declarative knowledge, retrieval) in order to enable anticipation and mental simulation.


On the other hand, the following features indicate low rationality and therefore unfreedom on the person dimension: psychiatric disorders, primarily those which impair volition, motivation and self-regulation (e.g., obsessive compulsive disorder, addiction); low cognitive and self-regulatory skills; cutbacks to memory functions; and insufficient knowledge about the self, its needs, and the environment.

### 3.5. Assessing the Functional Freedom in a Decision

To assess the capacity of functional freedom in a decision episode, three dimensional parameters must be considered: the underdetermination of the structure, the rationality of the decision-maker, and the amount of reflective processing. In decisions with very high functional freedom, the structure is characterized by diverse, important and conflicting options, which are considered by an individual with an elaborate self-concept and value system, yielding high cognitive and self-regulatory skills, by using a reflective, inferential and argument-driven process.

Consequently, it can be concluded that decisions with high functional freedom take longer until a choice is reached, are deeply elaborated (possibly manifested by profound retrospective justifications), show a low predictability of outcome, and a high subjective uncertainty of the decision-maker. By contrast, functional unfree decisions have a dominance structure or very few available options, are marked by intuitive, unconscious processing, and do not demand high cognitive and self-regulatory skills or an elaborate system of attitudes and values on the part of the decision-maker. Hence, functional unfree decisions are fast, effortless, dependent on intuition and routines rather than deliberation, and cause no uncertainty.

#### 3.5.1. What Constrains Functional Freedom?

Functional freedom is constrained by any parameter that changes the value of the three dimensions toward the “unfree” direction (i.e., unconscious process, determined structure, irrational person). Examples of constraints on functional freedom are inner processes that reduce or diminish rationality and the capacity for reflective processing, such as cognitive biases, sleep deprivation, uncontrollable affects (e.g., fear, rage), or strong distracting impulses (e.g., hunger). A good example of a structural constraint is blackmail. Blackmail manipulates a strong need of a victim (e.g., self-esteem, safety of the family) to threaten him/her (e.g., publish embarrassing information, threatening family) and restricts the victim to a limited choice-set. This set then contains one dominant option—the action that is instrumental to the blackmailer—which would exclusively *not* threaten the need in question (“do x for us and your family is secure, else…”). A strongly determined choice would result as the decision has then an externally induced dominance-structure.

As Fromm (1964) and Cranach (1996) assumed, one’s own behavior too might reduce the freedom of subsequent decisions. Unwillingness to learn from decisions and reluctance to use rational abilities likely decrease the mental repertoire in the long run. To illustrate, if someone is deciding to alleviate the burden of choice by choosing always to flip a coin from now on, he is (1) likely to forget or diminish competencies and methods of thinking and deciding, and (2) unlikely to be granted responsibility from others. Situations with high responsibility in turn are likely to hold high underdetermination. Consequently, people are able to restrict their functional freedom *themselves* over time.

#### 3.5.2. Characteristics of the Dimensions

Some closing remarks on the nature of the described dimensions: First, we emphasize that Figure 1 does not imply that the conditions of functional freedom are independent; the three dimensions are most likely correlated. For example, it is plausible to assume that high underdetermination and uncertainty tend to trigger a reflective processing style [45]. If so, it might be possible to reduce the structural dimension of underdetermination to the dimension of self-reflective process. However, empirical data on this question are inconsistent. Some studies show that decision conflicts may lead to avoidance or superficial processing [39,52], while others imply that complex choices elicit reflective processing [53]. In light of the unclear evidence, it is hard to reliably quantify the direction and magnitude of possible dependencies between the dimensions. Therefore, reductions would be unsubstantiated and it seems beneficial to assume three separate dimensions for now. More work and data is needed to differentiate the possible dependencies, especially how personality might affect the dimension values of structure and process (e.g., high tolerance of ambiguity might lead to an open embrace of high underdetermination).

It is yet difficult to determine concrete numerical values for the three dimensions. However, we can posit that the attribution of discrete metric values is possible because many of the indicators are already conceptualized as variables in quantitative research. Thus, to derive a metric level, one could aggregate indicators that are convertible as continuous variables (e.g., difference scores of preferences, severity ratings of psychiatric diagnoses) to a dimension value. A concrete example is suggested at the end of this paper, in Section 6.

The dimensions are understood as compensatory. That is, the overall functional freedom of a decision episode is assumed to be a weighted sum of the values on each dimension (i.e., underdetermination, rationality, reflection). We argue that an additive, compensatory function (i.e., a + b + c) makes more sense than, for instance, a multiplication of the dimension values (i.e., a × b × c). If the model was multiplicative, then very low values on *one* dimension, whether represented by negative values or values near zero, would always result in an overall negative or very low freedom. We believe it more reasonable to assume that the overall freedom of a decision is somewhat robust, in that it can make up for shortcomings in one dimension by growth in another (i.e., leading to a variety of differently free decisions). For example, decision structures with a low, “determined” value could be compensated for by an increase of freedom on the process dimension via increased reflection, cognitive restructuring, or the generation of new perspectives. A case supporting this claim was provided by Frankl, who described his adaptation to determining internment through the practice of self-reflection (e.g., his decision *not* to hate his guards, despite their cruelty, enabled him to react differently, such as to forgive) [29].

To be clear, every analysis of functional freedom is a “here and now” analysis that pertains to a concrete decision episode. Though it is relevant to further examine the relation of functional freedom to the freedom of action, and its development in subsequent stages of volition in the medium-term, for instance [54], this is beyond the scope of this paper.

We do believe that a comprehensive project on the psychology of freedom must link both areas: the psychology of motivation (i.e., decisions and goal setting) and the psychology of volition (i.e., goal implementation, action, and motor behavior). However, as the will (or intention) *within* actions is not the same as the will *before* actions [3,32], both areas have to be first understood in their specifics. The focus of this paper is on the conception of freedom in decision-making. To expand the approach to the freedom of action would entail many additional definitions and differentiations (e.g., stages of intention) [11,53,55]. We lack the room for it here, but see it as a next step. Recent research on the interaction between decision complexity and the capacity to abort and veto an action has already demonstrated the great potential for intriguing data when integrating decisions and volition [56].

## 4. Hypotheses, Benefits and Implications

Within this section, we will outline possible research avenues and hypotheses. Following, we illustrate adaptive benefits of functional freedom in decisions, as well as broader implications for research on freedom.

### 4.1. Hypotheses and Preliminary Support

#### 4.1.1. Subjective Freedom

The model of functional freedom enables a systematic test of the subjective, phenomenological condition of decision freedom, whose relations to ontological freedom remained long untested [57]. When do people feel free during decision-making, and do these feelings coincide with the theory? This question is relevant because intuitively we see a possible mismatch between the theoretical conditions of freedom described above and subjective experience [57]. Commonly, “freedom” seems associated with autonomy, high-mobility, fun, and the absence of obstacles. The according subjective representation, which people maintain and retrieve when they assess their freedom, is presumably of positive valence and rather refers to the freedom of action (i.e., doing what one wants) than to functional freedom (i.e., complex choices, conflict, and uncertainty). A recent study has supported this divergence between theoretical and subjective freedom, by showing that ratings of experienced freedom followed an outcome model (i.e., freedom predicted by easy choices and good consequences), but not a process model (i.e., freedom predicted by complex choices and underdetermination) [58]. This result demonstrated that, as supposed, *being* free and *feeling* free may not quite refer to the same conditions [58,59].

What significance do such findings bear for the model? Results on a divergence between subjective judgments and normative models are generally expedient, because they point to the need of an explanation and generate further research. Studies on agency or casuistic reports of free will indeed show similar interesting results, while they suggest negative [60], as well as positive [5,61] effects of conflict on phenomenological reports. We thus conclude that, for now, findings of divergence have neither confirming nor falsifying status for the model of functional freedom. First, functional freedom is defined as a capacity, which must not necessarily include a direct representation as “free”. Second, the conditions under which subjective and theoretical freedom *do* or *do not* correspond have yet to be clearly pinpointed and comprehensively determined. A hypothesis we pursue at the moment is that perceived competence represents a necessary boundary condition for a match of functional and subjective freedom.

#### 4.1.2. Underdetermination

Several predictions can be derived from decisions with high underdetermination. First of all, the choice behavior should become significantly less predictable (see also the suggestion on neuroscience below), as indicated by balanced proportions among options (e.g., 50% vs. 50% with two options) and a high subjective uncertainty about what to choose within the decision.

Furthermore, underdetermination should be positively associated with behavioral phenomena such as “variety seeking” [62]. Variety seeking describes a tendency to self-express one’s individuality by altering or varying one’s choices. It can be assumed that it would be easier for an individual to express himself via choice in underdetermined decisions because determined decisions feature a dominant option or default options that nearly everybody would choose. Moreover, if the seeking of variety is a dispositional tendency in which people differ, we can predict that they should also differ in their acceptance and evaluation of underdetermined choices. Underdetermination in decision-making is demanding on cognitive resources and holds ambivalence. Thus, traits such as variety seeking or the need for closure [63] might influence how people appraise underdetermined decisions, and hence how likely it is that they frequent those decisions and thus exert functional freedom. In sum, the psychological handling of underdetermination is likely prone to personal variance that should be investigated [44].

Given the diverging results already obtained with subjective freedom [58], the question arises whether people even *recognize* features of underdetermination (i.e., epistemic openness of choice) in a decision conflict. In a recent study, we tested the hypothesis that with increasing conflict, participants would judge the decision as more open, less predictable, and more self-determined [64]. A decision structure yielding a dominant option should accordingly be judged as more predetermined and predictable. The results supported these hypotheses: decisions with increasing conflict were judged as less determined, more self-determined (i.e., something that the person had to figure out by herself), and more open and unpredictable [64]. However, these judgments were again accompanied by a decrease in the feeling of freedom, which showed that parameters of epistemic openness do not seem to inform experienced freedom.

#### 4.1.3. Self-Reflection

It can be hypothesized that increased reflection during the decision process would lead to an enhanced congruity between own needs and actual goals, especially when taking foregoing decisions and events into account. There is some support that reflecting actively and constructively on own wants and experiences leads to more rewarding goal attainment via higher well-being [65] or an increased performance [66]. A related promising line of research could be to test the influence of conscious consideration on the quality, strength and neurological footprint of intentions [12].

Further research could test directly if decisions that are processed without conscious reflection are more suggestible to external and/or persuasive stimuli, and if resulting actions rather adopt the character of reactive behavioral responses than that of internal actions. For example, let us assume that one activates a certain motive (e.g., to reduce weight) experimentally. When reflection is then systematically inhibited, it should be easier to lead the participants towards choices and actions contradicting this need, as against when reflection is systematically endorsed. In parallel, a possibly positive impact of higher reflection on feelings of control and autonomy should be tested. Such research questions could well be tested within the framework of consumer choices [67].

#### 4.1.4. Rationality

As mentioned in Section 4.1.2., it can be assumed that the readiness to engage in underdetermined decisions depends on certain dispositions, such as the tolerance of ambiguity [68,69]. Accordingly, it is worthwhile to search for interactions of manipulations in process, such as inhibiting versus endorsing reflection during underdetermined choice, and traits that relate to rational decision-making (e.g., rational decision style) [70]. A higher working memory capacity may or may not increase the effect of conscious reflection on decision-related performance and well-being, for instance.

Functional freedom also permits predictions in the clinical domain. Negative, clinical indicators of rationality (e.g., disorders of volition, such as addiction or depression) should, for instance, relate systematically to other indicators of theoretical freedom. To illustrate, does the successful therapy of volitional disorders also predict increased occurrence of reflective processing, as well as the mentioned higher congruency of needs and goals, which shows in well-being and long-term life satisfaction? Another outcome in this respect could be the degree to which the individual becomes capable of reasonable ethical choices. There is some evidence on this matter [71], but the idea to relate freedom to psychological functionality still has to be explored further.

#### 4.1.5. Neuropsychology

Recently, Soon and colleagues demonstrated that simple decisions can be partially predicted based on neurological activity measured by functional magnetic resonance imaging [72,73]. Within this neurological paradigm, the hypothesis could be tested that increased epistemic openness would change those effects. The choice prediction should become more difficult (i.e., decreased accuracy, and a smaller time interval between prediction and actual decision) the higher the values of the parameters of functional freedom in the decision become. Specifically, significantly predictive neurological areas, as in [73], should emerge later and be harder to pinpoint when a decision is underdetermined and reflective, compared to dominant and spontaneous. Finding such systematic variance would stimulate ongoing discussions about the neurological reducibility of free will, as it would show that parameters of decision freedom would have a traceable effect on the neurological level as well.

### 4.2. Benefits and Implications

#### 4.2.1. What Is It Good for?—Adaptive Benefits of Functional Freedom

The topic of free will has been met with strong skepticism by some authors [74]. Others argue that investigations of free will are unscientific because it is a purely metaphysical or cultural construct [75]. We agree that it is right to be wary of metaphysical assumptions in psychology. However, our definition of decision freedom is neither a metaphysical construct nor an abstract cultural idea. We argue that the functional freedom in a decision has consequences, specifically adaptive ones. These arise from the utility that several authors ascribe to indicators, which are incorporated as freedom-endorsing in the model. To investigate the impact, mechanisms and scope of freedom in choices can open up new ways in which psychology can contribute to discussions about evolutionary adaptivity, ethics, and good life-goals.

First, several authors see an evolutionary benefit of free will that also generalizes to functional decision freedom; especially, mechanisms of rationality (e.g., self-control) facilitate the living within a culture [9,32], which itself is of great selection value [76]. The ability to follow rules, to control impulses and habits, and to adapt rules in a flexible manner is indispensable for the coherence of large cultural systems. Moreover, the processes of reflection and complex information integration facilitate exploration and learning behaviors, which replace rigid stimulus-response patterns, and thus likely create evolutional advantages [9,11,40,77]. A possible beneficial consequence of higher functional freedom in the form of underdetermination may also rest in an increased sense of agency or control [42]. Some evidence has shown a positive relation between larger choice sets and indicators of agency [78]. Although the relation between functional freedom and perceived control is probably complex and prone to moderation (Section 4.1.1.), it can be argued that attaining high freedom fosters the central function of control striving [79].

Second, reflective and hence free consideration probably enhances the congruence between self-needs and actual doing. For instance, Sheldon and Kasser demonstrated that progress in goal attainment is more rewarding in terms of well-being if these goals correspond to actual needs of the self [64]. Accordingly, high functional freedom in terms of self-reflection may endorse the long-term life satisfaction of people, as people are less likely to follow externally induced “inadequate ideas”, but rather needs that they really care about [20,21]. They might as well profit from the positive motivational effects of higher autonomy [8].

Third, there is also an ethical benefit. Strictly speaking, a link between freedom and western ethics can only be made under the assumption that the value system of the person contains values such as secular humanism. Given this condition, a combination of reflection, insight into own needs and abilities, and knowledge about the world might increase the “adeptness to act ethically” [20,21,30]. As functional freedom entails these processes, it can promote ethically adept decision-making. For example, the humanistic value not to judge other people by their racial background can, once reflected upon, have a greater impact on the generation of intentions (e.g., selection of job candidates) and the control of actions (e.g., treatment of co-workers). Even when viewed from a purely materialistic perspective (i.e., that discards values and the like), it is likewise assumed that a community whose members predominantly consider future consequences and evaluate on the importance of particular needs will prosper by showing a stable rate of adaptation to the environment [80].

Finally, we note that although we believe functional freedom to be beneficial to human capacity, it should not be misconceived as the only possible form of freedom. Functional freedom is a theory about how, beyond actions, freedom can be conceptualized and operationalized psychologically. This theory can be endorsed as an ideal, and can be pursued. On the other hand, one can consciously decide against this idea of freedom, perhaps because it relies too much on effort and process features, and instead rely on intuition and simple need satisfaction. The difference between functionally free and unfree decisions lies in motivational parameters that entail rather indirect and distal consequences. It is the task of people and societies, but not science, to decide which of those consequences are preferable, as it is the task of the society to determine the meaning and value of the concept of “freedom”. We presented arguments for an adaptiveness of functional freedom. Given the prominence of skepticism on the issue of free will and freedom [81], it is essential to demonstrate this utility in order to enable society to make an informed judgment about which notion of freedom to strive for.

#### 4.2.2. Functional Freedom Can Be Lost and Attained

Our model contradicts the common belief that the amount of human freedom is mainly set (e.g., by nature or political systems), and that free will or freedom is either given or not given. Functional freedom is attainable, and human beings have the capability to increase or decrease it by themselves.

A person who is striving for factors that are positively related to the three dimensions, such as education, meditation techniques, and situations with conflicting goals or high responsibility, is likewise increasing her capacity to make functional free decisions in the long run. Similarly, a person reading this article could decide *against* the attainment of functional freedom for some reason, and rely only on intuition instead. This person would then likely follow a different notion of freedom and would, in the frame of our model, be unfree afterwards. Finally, a person is also capable of restricting herself *unknowingly* by actively suppressing and avoiding hard choices, or by refusing to learn from and reflect on past actions and their conditions.

This implication underlines the great importance to: (1) actively consider one’s own position toward the concept of being free, and (2) make efforts to alter one’s own freedom in the desired direction.

#### 4.2.3. Implications for Research on Human Freedom

For theoretical analyses of the degree of freedom in certain psychological contexts, the model can help to categorize causal factors into either freedom-increasing or -decreasing. If we see functional freedom as a variable with adaptive benefits, the analysis of its antecedents becomes important, as we may want to increase the level of freedom in individuals. More importantly, we can identify factors that decrease decision freedom. Blackmail as an example of a freedom-decreasing cause was mentioned earlier. Another increase in determination would be achieved by depriving people of challenging, ambiguous choice situations. A non-structural example of a freedom-decreasing factor would be a depressive episode, which likely decreases rationality and processing capacities. Please note that the relationship between psychiatric disorders and functional freedom is not generally negative. For example, mild degrees of mania might facilitate creativity, and therefore increase capacities for a while [82].

On the other hand, a comprehensive education can be classified as a freedom-increasing factor because it is likely to increase rationality. Specifically, to learn and practice sophisticated methods of thinking would increase the share of critical self-reflection in everyday processing [83]. In general, the model permits specific predictions about parameters that will change the freedom of persons, which could also be changed by themselves.

In terms of classifying existing research, the model can likewise assist. Research on free will from recent decades has mostly been interpreted as an attempt to disprove the concept (e.g., the Libet-paradigm being the most famous) [72,84,85,86]. These intriguing studies share a common thread in their interpretation: conscious decisions are preceded, or caused, by unconscious brain activity. This means that decisions are in fact made before they enter awareness. Moreover, measuring the relevant patterns of brain activation enables one to predict the contingent processes up to 10 s before the decision enters awareness [72]. However, these data do not challenge the concept of functional freedom. The cited results [72,85,87,88] are largely restricted to *volitional* processes (i.e., monitoring of motor intentions, and action awareness) and simple actions (e.g., button presses), labeled as “bodily behavior” by Tuomela [32]. Thus, they convey interesting insights about the nature of volition, and the studies of Soon and colleagues especially represent a great methodical advance [72,73]. However, these data apply to the freedom of action, not to functional freedom of decision-making. Research on functional freedom must focus on the *motivational* processes and aspects of conscious decision-making that precede motor intentions (e.g., the decision to follow the instructions of the experiment, the pondering on a plan of how to time trial-responses, etc.). Only few studies have dealt with this domain so far, but there is promising work on the forming of conscious intentions, action vetos [11], and the role of diversity in stimulus input [12]. It is time to focus on the positive conditions of conscious will and freedom, and to determine their scope.

## 5. Concluding Remarks on Freedom as a Variable

Is the model of functional freedom at odds with a deterministic framework? No, because its extent is clearly limited by natural factors, and the conditions of freedom themselves (self-reflection, underdetermination, and rationality) are the object of determination by mental, biological and social factors. We can thus bypass the compatibility debate [4] by concluding that functional freedom is a psychological model of decision freedom.

Along similar lines, one might criticize the central role that conscious processing takes in the model, as it was repeatedly the target of skepticism [19,86,89,90,91]. While we believe that it is correct to scrutinize the impact and scope of conscious processing, critics must remain cautious not to exaggerate the role of unconscious processes. This holds especially for strong conclusions, such as that conscious processing is epiphenomenal. Many complex mechanisms, such as problem solving, depend on a continuous interplay of conscious and unconscious thought [22,24]. A broad body of literature demonstrates that conscious processing is involved in the guidance of decision-making, belief formation, and the coordination of executive and unconscious processes, as well as the implementation of logical and abstract reasoning [22,38,92,93,94]. These roles of conscious thought are perfectly consonant with the model of functional freedom. In sum, there is evidence that the processes associated with functional freedom (e.g., reflection, argumentation, mental simulation) not only require consciousness, but also have a discernible psychological impact.

## 6. Open Challenges

The most pressing task is to link the three dimensions to concrete indicators with valid numerical values. The following illustrates an according approach for the dimension of underdetermination. Underdetermination in a prototypical decision could be assessed, and possibly aggregated by: (1) the number of options; (2) a multiplicative term of number and absolute utilities of the options, as an indirect measure of the importance of the decision; (3) a direct measure of importance (e.g., via ratings); (4) a sum of the difference scores between ratings of preferences in the decision (i.e., a value near zero would indicate high underdetermination); and (5) a measure of familiarity with the decision problem (e.g., via a rating or, indirectly, by assessing prior experiences or reaction times for similar and dissimilar problems). This represents a preliminary attempt of how to allocate numerical and empirically tangible values to the dimensions. Given that all three dimensions can be assessed along those lines, prototypical examples of decisions with varying levels of functional freedom could be constructed. These prototypes could then serve as a basis for comparison, further theoretical elaboration and empirical operationalization of the functional freedom in decisions, and to relate the indicators to broader outcomes.

One of such outcomes is the attributed responsibility to actions, may that be in moral or legal situations. Responsibility is tightly connected to the idea of free control [11,47]. It is not a criterion of free will, but an often implied consequence. Thus, the concept of freedom or free will can serve to justify responsibility for actions and decisions. Commonly, an openness in choice and controllable, autonomous action are seen as prerequisites to ascribe responsibility and hence justify blame and punishment. A high functional freedom on all three dimensions can legitimize this type of responsibility as well: the more underdetermined (hence open), reflective and rational a decision, the larger the share of the decider in the outcome. This positive relation between functional freedom and responsibility becomes evident when we consider mitigating circumstances of responsibility, such as insanity, incapacity or blackmail [95]. These are factors that also decrease functional freedom. What remains open for future research is to analyze and investigate the *weight* of each dimension on responsibility judgments (e.g., does a determined choice structure outweigh the conscious and deliberate decision to kill someone?), and their possible interactions. Such an analysis would represent a possibly fruitful connection of psychology to the philosophy of law.

To sum up, via the model, decision freedom becomes available as a parameter for empirical psychological research. We outlined preliminary hypotheses as well as some support, and argued for adaptive benefits and a scope that could be defined, in which freedom in decision-making makes a difference as a parameter. In the end, what matters is that we engage productively with the topic of human freedom and that psychology takes its place among the disciplines already engaged with this intriguing subject.

## Figures and Tables

**Figure 1 behavsci-07-00041-f001:**
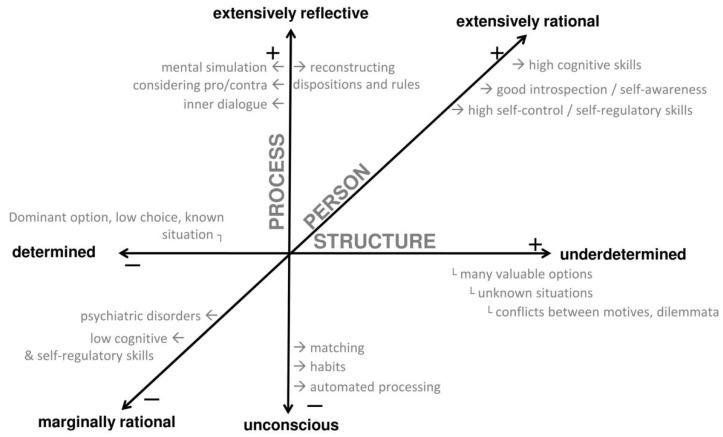
Dimensions of functional decision freedom. “+” and “−” denote higher and lower freedom, respectively, on each dimension.

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
