# Peer review of "Functional Freedom: A Psychological Model of Freedom in Decision-Making"

_behavsci, 2017, doi:10.3390/bs7030041_

Round 1

Reviewer 1 Report

Functional freedom:  A psychological model of freedom in decision-making

Authors:  Stephen Lau & Anette Hiemisch

The manuscript proposes a model of decision-making that includes a sense of freedom.  Overall, I like the model and it certainly expands on other decision-making models.  However, I feel some empirical data are needed to support the model.  It would be important to see how the three attributes interact and contribute to the functional freedom.     

Here are additional thoughts:

Line 36:  The manuscript briefly mentioned theories but does not elaborate on what they are.  It would strengthen the paper if each theory could be described a bit more.  Mentioning of why these models do not fully address the decision-making process would be a vital motivation for a new model (i.e., functional freedom). 

Line 63:  Functional freedom is defined here as the ability to will and regulate action.  It also implies consciousness.  This would be something similar to the idea that consciousness is needed to modify or cancel behavior.  The authors may wish to include work on the “free won’t” or “veto” window originally coined by Libet, and have been addressed by several other authors (Libet, 1985; Schultze-Kraft, 2016; Isham et al., 2017). The Isham et al. paper looks at automatic and deliberative decision making.  Perhaps the authors could consider how their data fit into the model. 

Line 95:  Christof Koch once stated that consciousness does not require “self-reflection.”  Here the manuscript implies consciousness and self-reflection go together.  It is minor, but the authors may wish to address this. Or, perhaps try to use a different term for self-reflection.

Line 137:  The dimensional indicators for self-refection are listed here. However, it seems that these indicators can be both objectively and subjectively measured.  For instance, one could measure objectively the focus of attention by looking at brain activity.  Subjectively, would could simply ask the observer of their attentional level.  If the two are incongruent, which is the more appropriate to determine functional freedom?  If subjective, would that suggest that functional freedom is epiphenomenal?

Line 166:  Work by Barlas & Obhi, 2013 demonstrated an increase in the sense of agency when more choices or options are available to choose from. It seems relevant for a brief discussion linking the sense of agency and functional freedom somewhere in the manuscript.  The authors potentially could offer the sense of agency as a benefit of functional freedom. 

Line 552:  I appreciate the implications of functional freedom especially on the perspective that it can be lost and attained.  

Author Response

First, we would like to thank the anonymous reviewers for their comments, the helpful suggestions and the interesting points of discussion. We will address their comments below, which we will also paraphrase as we understood them.

Reviewer #1

1. I feel some empirical data are needed to support the model. It would be important to see how the three dimensions interact and contribute to overall functional freedom.

Reviewer 1 is making a very comprehensible and good point here. However, we are currently not able to deliver the requested data in this extent, as this would require a complete new research program with presumably 6 - 12 experimental studies. If one wants not only to test predictions for the indicators of each dimension, but also examine interactions or dependencies between them, this estimate ends on the larger side. The aim of this paper was to lay down the conceptual foundation for exactly that, and we agree fully that empirical approaches to the dimensions are needed in future. We tried to compromise by delineating possible hypotheses and research questions in section 4.1. We also report some preliminary empirical results on the experience of freedom and the perception of underdetermination, which tested aspects of the model (i.e., the process and the structural dimensions). However, these results lead to their own distinct questions, as people seem to recognize underdetermination correctly, but they do not associate this with a subjective feeling of freedom (see 4.1.1, line 457f. and 4.1.2, line 484f.). Direct tests of the model will come but they will also need more conceptual work in terms of finding concrete indicators for lab experiments (see our example at line 719f.) and, even more important, in terms of defining outcomes that are indicative of more or less freedom.

2. At line 36 the authors briefly mention literature and theories, but do not elaborate on them. It would strengthen the paper when the authors would describe these a bit more and thereby flesh out their motivation to add to these by a new model about the decision making process.

This is a good suggestion, we might have swept through the recent literature there too fast. The point of these citations is to show how active and lively psychology’s take on the topic of free will has become (after virtually ignoring it until the late 1990s), and that a variety of results and positions have been established. We have added more elaboration on each citation and also stated more clearly how we want to build on that to contribute a further step in this endeavor (see line 43f.).

3. Line 63, the definition as the ability to will and regulate action is similar to the idea of a conscious behavioral veto as marker of free will (following Libet, 1985). Recent results have examined this veto-capacity and also connected it to differential forms of decision making (see Isham et al., 2017). Please consider how this data can fit into the model.

The veto capacity is an interesting phenomenon and we are grateful for the suggested literature (especially the intriguing work of Isham and colleagues, which was prior unknown to us). In considering this capacity (and the data on it), we have to carefully analyze its location and logical relations to the concept of decision freedom. Given the suggested literature and other approaches (Haggard, 2008; Brass & Haggard, 2007) we can locate the veto capacity rather late in the action process, that is, after decisions about more abstract and general forms of intentions (the what, why and when) are already made. Next step would be to initiate motor processes (see also Pacherie, 2007, motor intentions in distinction from future and present intentions). This close proximity to goal directed activity and motor responses suggests a deeper relation to volitional action control (i.e., effective intentions, bodily behavior) than motivational action control (i.e., goal setting and need evaluation, see Tuomela, 1977; Gadenne, 2004). The function of the veto presumably is to “double check” and compare the planned output (e.g., motor efferences) with the output created in prior deliberation (i.e., the decision and guiding “long-range-intentions”, see Haggard, 2008). Therefore, we can make two conclusions. First, the veto capacity is not an inherent part of functional decision freedom, as this concept is applicable to decision-making and the generation of rather abstract, distal intentions (e.g., Mele, 2009). When we are thinking about how and why to choose, we do not yet need a veto. Therefore, we will not model it here. Second, the veto capacity is nevertheless related to functional freedom and the freedom of action, because, as has been nicely stated by reviewer 1, it represents a conscious influence, possibly in form of a decision, on action control. We see this relation as follows: When an intention has been generated functionally free it can remain as mental representation (e.g., a goal) or be further processed to lead to concrete actions. A free intention that is implemented as overt action enters the domain of the freedom of action (i.e., the freedom to do what one wants). In our view the degree of the freedom of action for a certain intention-implementation process is characterized by two factors – the congruency with the originally free decision (do my actions really lead to and represent what I decided?) and the degree of constraints or success-enhancing factors (do I achieve what I want?). The veto capacity can be seen as a mechanism to achieve the first part – by consciously “going back” to the original decision and comparing the actual plan or motor output to this mental representation. If no congruency is perceived, people can abort the action (as Schulze-Kraft et al., 2016 and Brass & Haggard, 2007 demonstrate), and adapt. This would secure a high freedom of the action, as it increases the chances for the originally made decision to be implemented. However, the order is clear -  the veto window always follows after a decision has been made. Therefore, we could see it as a criterion of the demarcation between conscious action control, where decisions can still be made, and motor behavior. An exact demarcation of these two phases was left open by several authors (Tuomela, 1977; also Haggard, 2008; Pacherie, 2007) and is thus very helpful. Isham and colleagues even show that this criterion can vary considerably in terms of the time window, depending on the nature of foregoing processes. This is very intriguing work that helps to bind the volitional (action) to the motivational (decision) realm.

To sum up, we cannot include the veto capacity into our model of decision freedom, as it is a subsequent process. To delve into our thoughts about further processing of a decision and volitional versus motivational aspects of freedom would lead to far and dilute the present paper. However, we agree about the relevance and plan to work more on this. We also have included some additional thoughts and prospects about this issue at line 430f.

4. Christof Koch stated that consciousness does not require self-reflection. The manuscript, however, implies that both go together. The authors may wish to address this, or try to use a different term for self-reflection.

This is an interesting point of discussion, as the term of self-reflection is associated with a lot of connotations (mostly from folk psychology) and is not consistently defined within psychology. We will answer to this comment by first clarifying the role of reflection in consciousness as we understand it and why it has to go together with consciousness, and second by arguing that Koch’s position seems quite well in agreement with our view. Based on these reflections, we also clarified the relation between consciousness and self-reflection more, please see line 157.

In short, consciousness is necessary but not sufficient for self-reflection. That is, reflection can only exist within consciousness, but of course consciousness must not entail self-reflection. Phenomenal awareness, for example, is consciousness without controlled processes, reflections or conscious thought about oneself (Gadenne, 1996; also Baumeister & Masicampo, 2010). However, consciousness frequently goes along with and is thus strongly characterized by controlled, System 2-like processes. We thus understand reflective processes as criterions for and applications of consciousness in decisions. And these applications are necessary for the types of decision control we propose as freedom endorsing – insight, analytic argumentation, simulation, perspective taking and so on. Evidence shows that putting load on conscious processes impairs these analytical capacities (DeWall, Baumeister, & Masicampo, 2008), while it would probably leave the conscious awareness completely intact. Admittedly, a lot of sophisticated processes, like pattern recognition or attitude forming, are working properly without much access to System 2 and reflective processing. But these processes are also not the pivotal point when it comes to the question of re-determining exterior influences and re-shaping thoughts, attitudes, preferences and ultimately behaviors in accordance with the needs and values of the self. For this we need specific modes of thinking, describable (in our view) so far only by conscious thought and reflection (see also Johnson-Laird, 1988). And as functional freedom always assesses an individual in a concrete decision episode, those thoughts are chiefly centered or referenced to the self. This is why we use self-reflection as label (instead of ‘reflection’ which could also mean to think about entities unrelated to the self). We are aware of and agree that ‘self-reflection’ is still a somewhat problematic term, due to its inconsistent definition and various connotations. But we did not find a better one yet.

Of course, to be consciously aware of influences, desires, entities is an indispensible part of freedom, this is where we meet Koch’s (2009, 2012) interesting position. While he refutes Cartesian free will (2009), which we fully agree on, he seems to see uncertainty and stochastic randomness as more natural criterions for it (Koch, 2009, 2012). He also emphasizes the need to be aware of own unconscious motivations and desires by reflecting about own actions and feelings (Koch, 2012, p. 27). These assumptions are well compatible to our model, as uncertainty and unpredictability are mapped by the condition of underdetermination (i.e., decision structures with ambiguity or conflict) whereas the reflection on own mental states and actions is part of the reflective process dimension. The result, an increased wisdom, is not only briefly mentioned in Koch’s 2012 article but also an indicator of the rationality dimension of functional freedom.

 5. Line 137, the dimensional indicators of self-reflection. These can be measured both subjectively and objectively, for example, self reports of attention versus objective measures (eye-tracking, reaction times, brain activity). Which of both types is more appropriate to determine functional freedom, especially when they are incongruent? Subjective criterions as basis could mean that functional freedom is epiphenomenal, couldn’t it?

This is a very insightful question. Our direct answer would be that the objective measures are always more appropriate to indicate functional freedom. Not only because functional freedom should rest on firm and reproducible experimental parameters with clear behavioral impact, but also because recent results have already shown that the subjective experience (of volition or freedom) can be an ambiguous source of data (see sections 4.1.1. and 4.1.2.). Subjective responses and experiences, especially toward decisions, can be seen as descriptive and biased (Baron, 2008). Therefore, they might tell us interesting facts about what the people think and belief but this is not necessarily to be equated with what is going on (i.e., normative or ontological models). Westcott cautioned to always differentiate between the ontological condition of being free and the experience of freedom (1988, p. 136). And while empirical data on the experience of freedom in decisions seems to divert from predictions of functional freedom (Lau et al., 2015), it is a distinct category with yet not strong and consistent enough findings to disconfirm ontological claims (see also Browning, 1964). We propose the direction should be to first establish the ontological model from which then to derive and test further predictions about subjective and descriptive measures. We also generally encourage the use of objectifiable and quantifiable measures wherever possible to test consequences of differences in functional freedom directly (see section 4.1). And as the model is chiefly based on psychological mechanisms with a behavioral impact and evolutionary values (attention, memory, self-control) we can safely assume that functional freedom is not epiphenomenal.

6. Work by Barlas & Orbi (2013) indicated an increase in the sense of agency when more options or choices are available. It seems relevant to link the sense of agency and functional freedom somewhere in the manuscript. One could for example offer the sense of agency as a benefit of functional freedom.

This is a welcome suggestion and we have made according amendments in the section (see line 573). We would also like to devise caution here, as agency (and the sense of it) are often linked to free will, which again seems to establish a connection to decision freedom. The sense of agency, however, is tapping at experiences that are accompanying actions but not necessarily choice per se (Browning, 1964). Smooth actions without any conflicts (and thus need for deliberative thinking) can also yield a high degree of agency (see Wenke et al., 2010; line 471). Moreover, it was argued (Schwartz, 2000) and shown (Lau et al., 2015) that more choice options do not generally increase subjective freedom, but can indeed even decrease it. Therefore, it seems premature to assume a linear relationship between the amount of choice and subjective freedom or agency. Its effect might depend on more specific situational details and moderators, as for example a decrease in choice can again cause reactance-like effects (see Lau & Wenzel, 2014; Damen et al., 2014). As with the role of phenomenological freedom in general (see above, point 5), this remains a puzzling and worthwhile field of research. To conclude, we wholeheartedly agree with the notion that to have a sense of agency and believe in agentic control is certainly a benefit that comes from high freedom (it is also echoed by many authors, e.g., Cranach, 1996; Herrmann, 1996; Vollmer, 1975; Heckhausen). We think though that this effect is considerably moderated by competence variables (see line 480f.). This even fits to Haggard’s take on intentional binding (see 2005, 2008) as he believes it to be a necessary superstructure for effective agentic learning experiences.

References

Baron, J. (2008). Thinking and deciding (4th ed.). New York: Cambridge University Press.

Baumeister, R. F. (2008). Free will in scientific psychology. Perspectives on Psychological Science 3(1), 14-19.

Baumeister, R. F., & Masicampo, E. J. (2010). Conscious thought is for facilitating social and cultural interactions: How mental simulations serve the animal–culture interface. Psychological Review, 117(3), 945-971.

Brass, M., & Haggard, P. (2007). To Do or Not to Do: The Neural Signature of Self-Control. The Journal of Neuroscience, 27(34), 9141-9145.

Browning, D. (1964). The Feeling of Freedom. The Review of Metaphysics, 123-146.

Cranach, M. v. (1996). Handlungs-Entscheidungsfreiheit: ein sozialpsychologisches Modell [„Action-decision freedom: A social-psychological model.“]. In M. v. Cranach & K. Foppa (Eds.), Freiheit des Entscheidens und Handelns (pp. 253-283). Heidelberg: Asanger.

Damen, T. G., van Baaren, R. B., & Dijksterhuis, A. (2014). You should read this! Perceiving and acting upon action primes influences one's sense of agency. Journal of Experimental Social Psychology, 50, 21-26.

DeWall, C. N., Baumeister, R. F., & Masicampo, E. J. (2008). Evidence that logical reasoning depends on conscious processing. Consciousness and Cognition, 17, 628-645.

Gadenne, V. (1996). Bewusstsein, Kognition und Gehirn: Einführung in die Psychologie des Bewusstseins. Bern: Huber.

Gadenne, V. (2004). Philosophie der Psychologie. Bern: Huber.

Haggard, P. (2005). Conscious intention and motor cognition. TRENDS in Cognitive Sciences, 9(6), 290-295.

Haggard, P. (2008). Human volition: Towards a neuroscience of will. Nature neuroscience, 9, 934-946.

Herrmann, T. (1996). Willensfreiheit - eine nützliche Fiktion? [„Free will – A useful fiction?“]. In M. v. Cranach & K. Foppa (Eds.), Freiheit des Entscheidens und Handelns (pp. 56-69). Heidelberg: Asanger.

Johnson-Laird, P. N. (1988). The computer and the mind. Cambridge, Mass.: Harvard, University Press.

Koch, C. (2009). Free Will, Physics, Biology, and the Brain. In N. Murphy, G. R. Ellis & T. O’Connor (Eds.), Downward Causation and the Neurobiology of Free Will (pp. 31-52): Springer Berlin Heidelberg.

Koch, C. (2012). FINDING FREE WILL. (cover story). [Article]. Scientific American Mind, 23(2), 22-27.

Lau, S., & Wenzel, M. (2014). The effects of constrained autonomy and incentives on the experience of freedom in everyday decision-making. Philosophical Psychology, 1-13. doi: 10.1080/09515089.2014.951718

Lau, S., Hiemisch, A., & Baumeister, R. F. (2015). The experience of freedom in decisions – Questioning philosophical beliefs in favor of psychological determinants. Consciousness and Cognition, 33(0), 30-46.

Libet, B. (1985). Unconscious cerebral initiative and the role of conscious will in voluntary action. Behavioral And Brain Sciences, 8(4), 529-566.

Pacherie, E. (2007). The Sense of Control and the Sense of Agency. Psyche, 13(1), 1-30.

Schultze-Kraft, M., Birman, D., Rusconi, M., Allefeld, C., Görgen, K., Dähne, S., . . . Haynes, J.-D. (2016). The point of no return in vetoing self-initiated movements. Proceedings of the National Academy of Sciences, 113(4), 1080-1085.

Schwartz, B. (2000). Self-Determination. The Tyranny of Freedom. American Psychologist, 55(1), 79-88.

Spinoza, B. (1677/1988). Die Ethik. Leipzig: Reclam.

Stachowiak, H. (1973). Allgemeine Modelltheorie. Wien, New York: Springer-Verlag.

Tuomela, R. (1977). Human action and its explanation: A study on the philosophical foundations of psychology. Dordrecht, Holland & Boston, MA: D. Reidel Publishing.

Vollmer, G. (1975). Evolutionäre Erkenntnistheorie. Stuttgart: Hirzel.

Wenke, D., Fleming, S. M., & Haggard, P. (2010). Subliminal priming of actions influences sense of control over effects of action. Cognition, 115(1), 26-38.

Westcott, M. R. (1977). Free will: An exercise in metaphysical truth or psychological consequences. Canadian Psychological Review/Psychologie canadienne, 18(3), 249-263.

Westcott, M. R. (1988). The psychology of human freedom: A human science perspective and critique. New York: Springer.

Reviewer 2 Report

This paper is an attempt to outline a psychological model of decision freedom that increases with rationality, undeterminism, and reflection / conscious thought. The model appears potentially novel and the manuscript has some interesting ideas. For example, the distinction between freedom of action and that of motivational processes is interesting and worthy of discussion. The authors note that much modern psychological and neuroscientific discussions of volition are more an attempt to disprove free will than to understand the factors that contribute to increasing freedom. The authors view their work as an attempt of the latter kind. Unfortunately, like many other attempts to ascribe characteristics to increase freedom or free will, the effort falls short for various reasons elaborated below.

Major comments

·      The authors repeatedly state that their model is psychological. But it is not clear why. A typical psychological model is based on some empirical findings and results in additional, non-trivial empirical predictions that can be tested. The model here does relate to some factors that are studied in psychology, like rationality and reflection. And the authors suggest that some experiments could study correlations between dimensions of the model. But it is not clear based on what empirical data the model has been constructed (though some sources are sporadically mentioned). The authors do not suggest a way to test the model as a whole empirically. As such, naming the model “psychological” (especially already in the title) seems strange.

·      Another option could be that this model is theoretical, philosophical model, with its empirical verification left to other work. But it does not seem to be theoretically derived from first principles either. As such, it is unclear why the reader should accept some of the main tenets of the model, such as its separation into the specific axes that make it up. What is it that make these characteristics the critical ones on top of which to construct the model?

·      The main idea in the manuscript seems somewhat confused, amalgamating intuitions and theoretical ideas. In parts, it appears similar to the long-standing system 1 / system 2 distinctions. And the manuscript does not make it clear enough in what this model is different from previous ones.

·      The manuscript also seems to associate freedom with responsibility (e.g., lines 224, 235) without clarifying this relationship. At the same time, they suggest that the relation between freedom and responsibility is unclear (lines 607-611).

·      What is the advantage of “high functional freedom”? The authors note that it is not inherently better (e.g., 507-510). So why devote the manuscript to this? For instance, the authors note a few times that humans can decide for or against attaining more functional freedom (e.g., 527-529). Such a decision could seemingly be made freely according to their model. But they claim various advantages to high functional freedom (section 4.2.1). So, it seems that a rational person, if convinced by their argumentation, should opt for higher functional freedom. There seems to be a contradiction here.

·      The authors devote Section 4.2.1 to the adaptive benefits of functional freedom. But they seem to align it with free will and freedom in general, citing benefits that others associated with these terms. However, this seems to beg the question as it assumes an identification between what the model terms as freer and what other authors associated with freedom or freedom of will.

Minor comments

·      Lines 68-69: Undetermined decisions are deemed more free and easier decisions (associated with automated decisions) are deemed less free. I am not sure I agree. A choice between a $1 and $10 immediate reward seems both easy and determined (take the $10 now). But perhaps one’s freedom is precisely taking the $1 now to show one’s idealism or unconventionalism, to give just one example. So, appears to be a simplistic account of decision-making.

·      Line 130-131: The claim that recent studies associated freedom with conscious processing is not necessarily correct. It might further be that the question of whether conscious decision-making is freer than unconscious is not an empirical one. It seems to rely more on definitions of freedom and thus to be a more of a conceptual/theoretical one.

·      Some sentences are slightly convoluted. The paper would thus benefit from careful reading by the authors.

Author Response

First, we would like to thank the anonymous reviewers for their comments, the helpful suggestions and the interesting points of discussion. We will address their comments below, which we will also paraphrase as we understood them.

Reviewer #2

1.a The authors repeatedly state that their model is psychological, while it remains unclear why. Typical psychological models are based on empirical findings and enable non-trivial, testable predictions. The model here does relate to some factors studied in psychology and the authors suggest some experiments that could study correlations between the dimensions. It is not clear on hwat empirical data the model has been constructed. The authors do not suggest a way to test the model as a whole. As such, “psychological” seems strange.

We are thankful for this comment, as it gives us the opportunity to reflect on our project. It is also a tough one as the answer depends much on what one understands as a ‘psychological model’. We will try to address the reviewer’s concerns by: 1) discussing the definitive features of models in psychology, 2) arguing for a rationalistic, theoretical but distinctively psychological basis of our model, 3) stating the resulting aim again more clearly and 4) highlight which empirical predictions are possible at this point and which are not.

First of all, we question the assumption that a theoretical model has to be based on empirical results to be psychological. Models are generally simplified mappings of reality, they can vary greatly in terms of complexity, content and basis (Stachowiak, 1973). To be sure, modern psychological models are often empiristic (i.e., based on inductive inference from data) and follow certain paradigms. However, they can also be constructed rationalistic (i.e., derived conceptually from other theories or definitions) and examples can be found within older theories, like Lewin’s field theory approach, which was at first a reconstruction of physical field theories.

Our approach follows the rationalistic way. Our endeavor was to ultimately arrive at a psychological re-construction of the concept of free will, which translates to assumptions about high or low decision freedom, without any of the metaphysical ballast tied to philosophical free will (i.e., questions about determinism and compatibility, questions about Cartesian dualism and the soul). Our premises may not have been stated clear enough and we are thankful for this suggestion (see below, 1.b, and line 121). But apart from that the psychological and hence scientific nature is warranted, as the model builds on naturalistic philosophical and psychological literature, proposes only indicators that are accessible by experimental research (though some, as for instance wisdom, may be underrepresented still), and already allows to derive implications for other psychological constructs and to suggest research questions (e.g., line 544f.).

The central objective in this approach is to start with a comprehensive conceptual basis what exactly freedom in decisions is about, which components and parameters are involved and how we can define and confine states of freedom from states of unfreedom in a decision. This approach is following what other psychologists have requested before (see Westcott, 1977).

Concluding, we’d like to emphasize that in this conceptual stage of the model it is too early to derive predictions about the model as a whole. Functional freedom is a complex and abstract variable that can (for now) probably not be subsumed under one measure. This is why we try to encourage empirical research on its aspects first (see section 4.1.1 to 4.1.5.). Such results are also necessary to conceptually tackle the possible interdependencies between the dimensions (e.g., a positive relation between reflective process and rational decider), and only then become plans about a complete testing possible. However, we’d like to stress again that the primary objective of this paper is to lay a conceptual foundation.

1.b Another option is that this model is theoretical or philosophical. But it does not seem to be derived from first principles either. It is unclear why the reader should accept some of the main tenets of the model such as its separation into specific dimensions. What is it that make these characteristics the critical ones?

Thank you for this comment. Yes, we agree with the notion of a theoretical model (not a philosophical though). We may have been streamlining the paper too much and thereby lost sight of the logical premises that guided our conception of this model. We included a paragraph about this again, please see line 121f. To sum it up here: The model represents a compatibilistic approach to free will with the aim to reconstruct it as a psychological and hence ultimately empirical variable. This means that we accept weak conceptions of determinism (including chaos and probabilistic causation, see for example Koch, 2009) and see free will as a relative inner freedom that can be understood as human capacity. Our premises now rest on largely psychological approaches (Cranach, Herrmann, Dörner, Bieri, Johnson-Laird, Baumeister, Walter, Fromm etc.) that agree on this notion of freedom as a capacity and follow the philosophical assumptions of Spinoza (1677/1988), that is, freedom as rooted in the disposition to reflect on own motives and affects by the process of insight. The analysis of this literature has shown that although all authors converge on the central assumption of freedom as a capacity, different authors locate this capacity on different levels, and either highlight the process (e.g., Johnson-Laird, Baumeister, Bieri) or the individual (Fromm, Spinoza, Frankl) or the situational structure of the choice (Cranach, Herrmann). These are all valid points, and it thus occurred to us that a model of decision freedom must take all the dimensions into account on which decisions are examined by research – process, person, and structure.

2. The main idea in the manuscript seems to confuse intuitions and theoretical ideas. In parts, it appears similar to the dual process models of information processing (System1 versus System2). It does not make clear enough in what this model differs from previous ones.

We have tried to be clearer on the motivation and contribution of our model in the introduction (see line 43f.). The main difference (or contribution) of our model is that it tries to integrate the various different conceptions (see 1.b above) of naturalistic, psychological free will into one framework, and enables the assessment of high or low freedom in a decision episode as one outcome variable. As mentioned above, we also state our premises and approach for the modeling more explicitly. The perceived similarity of some of the model’s assumptions with the System1/2 debate probably rests on the statements concerning the process dimension. Self-reflective processing (i.e., free processing) of a decision shows a great overlap with the process normally studied under System 2, however, both are not the same. We make clearer now (see line 157) what we mean with self-reflection. It is a specific type of consciousness that is (in contrast to intuitive or unconscious processing) necessary to map freedom in a decision process.

3. The manuscript seems to associate freedom with responsibility without clarifying this relationship.

Thank you for this comment. In short, responsibility is not a criterion of free will (as sometimes purported) but a consequence of it, and thus also a common correlate. A person who is in possession of agentic and autonomous control over his choices and actions is responsible for these actions and can be ascribed blame, praise or punishment. This does also mean that there is a strong positive correlation between freedom and responsibility, which is no different for the case of functional freedom. We see that we did not sufficiently explicate this relation and we do so now (see line 732). We also qualify what is still unclear about the relation between functional freedom and responsibility.

4. What is the advantage of high functional freedom? The authors note several times that it is not per se better. So why devote the manuscript to this? They later mention several advantages to high functional freedom so that a rational person, if convinced, should opt for functional freedom in any case. There seems to be a contradiction between those two assertions.

This is a good point, as we can see the possible contradiction there too. We made according changes to the paper (see line 608f.).

The main reason for including the parts that are cautious about the normative status of functional freedom is that we did not want it to be understood as a singular model or conception of inner freedom (or free will) that rules out all others. We encountered a vast variety of conceptions and ideas about freedom and free will during our literature search, a lot of which are not even mentioned here or incorporated, as we follow the freedom-as-capacity school of thought. We would like to see our model at eye level with other approaches, and possibly competing, but not established as the-one-and-only. Furthermore, the ideas of freedom and free will are ultimately socio-cultural representations, and hence change with time or cultural background (Cranach, 1996). This is why we see it logically permissible that a reader can or cannot follow our definition of freedom as a decisional capacity, or instead see freedom as something revolving only around his desires and their satisfaction. In general, what makes freedom or free will, will always be to some degree a conceptual and not empirical question, as “freedom” is a genuinely human invention.

5. Section 4.2.1 is devoted to benefits of functional freedom. However, it conflates or not satisfactorily confines benefits of functional freedom from benefits of other conceptions of freedom or free will.

This is correct and we try now to be more clear in this respect. Of course, the crux is that the model of functional freedom is partly homomorphous with those conceptions of other authors, as it is build upon them (see 1.b above). This does also mean that the benefits that those authors describe for their conceptions generalize to functional freedom. However, we try to be more clear now in the references to the parameters and indicators of functional freedom (see line 566f.).

6. I am not sure I agree that underdetermined decisions are deemed more free and easier decisions (associated with automatization) are deemed less free. A choice between $1 now and $10 now seems both easy and determined. But perhaps one’s freedom is precisely to take the $1 to show one’s idealism, for example.

This is an interesting example, but we do not fully agree with the conclusion derived from it. The existence of underdetermination (versus determination) is defined as a conflict between preferences. Preferences are values that people assign idiosyncratically to certain choice options. It is therefore hard to find an example of an underdetermined or determined choice (as given above) that generalizes to all people in all cases. We have to assume that the value of the options, in this case money, is constant for all to assume a determination due to a dominant $10 option. Hence, for all people that value money in the way (money = important AND more = better) the decision above is clearly determined, they will most probably take the $10 without thinking twice. However, reviewer 2 is fully correct about the assumption that there is more to that in terms of freedom, and idealism (money = unimportant to me) is a nice idea. However, it only swaps the preference situation and the outcome for the example. When an individual assigns other, non-default values to those options, we have to take these into account and adjust our prediction. Please note that this would not change the status of a determined choice structure IF the ideal of this particular person is very strong (i.e., money is completely appalling to him). Then he would probably always choose the $1 without thinking twice. He would not be free in terms of underdetermination either. To construe an underdetermined conflict out of this, we would need a person who entertains this ideal AND is nevertheless in need of money. Then we could not safely predict if he would go for the $10 (to pay his bills) or for the $1 (to endorse his idealism). In any case, this is well accommodated by the theory and the assumptions of underdetermination.

7. The claim that recent studies associated freedom with consciousness (line 130) is not necessarily correct. The question if conscious decisions are freer than unconscious seems not an empirical one, rather a conceptual/theoretical definition.

We agree on the interpretation of that question as a primarily conceptual one. Throughout section 3.1.1 and 3.1.2. we also argue why we see conscious self-reflective processes as necessary part of a free decision process. It enables the kind of thought processes that make choices and behavior less rigid, less stimulus-response-like and give more activation to own needs and values in long-range intentions. We have changed the wording of the citation of the mentioned results to be more clear as well (see line 187).

References (in the complete response letter)

Baron, J. (2008). Thinking and deciding (4th ed.). New York: Cambridge University Press.

Baumeister, R. F. (2008). Free will in scientific psychology. Perspectives on Psychological Science 3(1), 14-19.

Baumeister, R. F., & Masicampo, E. J. (2010). Conscious thought is for facilitating social and cultural interactions: How mental simulations serve the animal–culture interface. Psychological Review, 117(3), 945-971.

Brass, M., & Haggard, P. (2007). To Do or Not to Do: The Neural Signature of Self-Control. The Journal of Neuroscience, 27(34), 9141-9145.

Browning, D. (1964). The Feeling of Freedom. The Review of Metaphysics, 123-146.

Cranach, M. v. (1996). Handlungs-Entscheidungsfreiheit: ein sozialpsychologisches Modell [„Action-decision freedom: A social-psychological model.“]. In M. v. Cranach & K. Foppa (Eds.), Freiheit des Entscheidens und Handelns (pp. 253-283). Heidelberg: Asanger.

Damen, T. G., van Baaren, R. B., & Dijksterhuis, A. (2014). You should read this! Perceiving and acting upon action primes influences one's sense of agency. Journal of Experimental Social Psychology, 50, 21-26.

DeWall, C. N., Baumeister, R. F., & Masicampo, E. J. (2008). Evidence that logical reasoning depends on conscious processing. Consciousness and Cognition, 17, 628-645.

Gadenne, V. (1996). Bewusstsein, Kognition und Gehirn: Einführung in die Psychologie des Bewusstseins. Bern: Huber.

Gadenne, V. (2004). Philosophie der Psychologie. Bern: Huber.

Haggard, P. (2005). Conscious intention and motor cognition. TRENDS in Cognitive Sciences, 9(6), 290-295.

Haggard, P. (2008). Human volition: Towards a neuroscience of will. Nature neuroscience, 9, 934-946.

Herrmann, T. (1996). Willensfreiheit - eine nützliche Fiktion? [„Free will – A useful fiction?“]. In M. v. Cranach & K. Foppa (Eds.), Freiheit des Entscheidens und Handelns (pp. 56-69). Heidelberg: Asanger.

Johnson-Laird, P. N. (1988). The computer and the mind. Cambridge, Mass.: Harvard, University Press.

Koch, C. (2009). Free Will, Physics, Biology, and the Brain. In N. Murphy, G. R. Ellis & T. O’Connor (Eds.), Downward Causation and the Neurobiology of Free Will (pp. 31-52): Springer Berlin Heidelberg.

Koch, C. (2012). FINDING FREE WILL. (cover story). [Article]. Scientific American Mind, 23(2), 22-27.

Lau, S., & Wenzel, M. (2014). The effects of constrained autonomy and incentives on the experience of freedom in everyday decision-making. Philosophical Psychology, 1-13. doi: 10.1080/09515089.2014.951718

Lau, S., Hiemisch, A., & Baumeister, R. F. (2015). The experience of freedom in decisions – Questioning philosophical beliefs in favor of psychological determinants. Consciousness and Cognition, 33(0), 30-46.

Libet, B. (1985). Unconscious cerebral initiative and the role of conscious will in voluntary action. Behavioral And Brain Sciences, 8(4), 529-566.

Pacherie, E. (2007). The Sense of Control and the Sense of Agency. Psyche, 13(1), 1-30.

Schultze-Kraft, M., Birman, D., Rusconi, M., Allefeld, C., Görgen, K., Dähne, S., . . . Haynes, J.-D. (2016). The point of no return in vetoing self-initiated movements. Proceedings of the National Academy of Sciences, 113(4), 1080-1085.

Schwartz, B. (2000). Self-Determination. The Tyranny of Freedom. American Psychologist, 55(1), 79-88.

Spinoza, B. (1677/1988). Die Ethik. Leipzig: Reclam.

Stachowiak, H. (1973). Allgemeine Modelltheorie. Wien, New York: Springer-Verlag.

Tuomela, R. (1977). Human action and its explanation: A study on the philosophical foundations of psychology. Dordrecht, Holland & Boston, MA: D. Reidel Publishing.

Vollmer, G. (1975). Evolutionäre Erkenntnistheorie. Stuttgart: Hirzel.

Wenke, D., Fleming, S. M., & Haggard, P. (2010). Subliminal priming of actions influences sense of control over effects of action. Cognition, 115(1), 26-38.

Westcott, M. R. (1977). Free will: An exercise in metaphysical truth or psychological consequences. Canadian Psychological Review/Psychologie canadienne, 18(3), 249-263.

Westcott, M. R. (1988). The psychology of human freedom: A human science perspective and critique. New York: Springer.

Round 2

Reviewer 1 Report

I have reviewed the revision and I believe the recent additions have significantly improved the manuscript.